# Cryopreservation of Human Spermatozoa: Functional, Molecular and Clinical Aspects

**DOI:** 10.3390/ijms24054656

**Published:** 2023-02-28

**Authors:** Lara Tamburrino, Giulia Traini, Arianna Marcellini, Linda Vignozzi, Elisabetta Baldi, Sara Marchiani

**Affiliations:** 1Andrology, Women’s Endocrinology and Gender Incongruence Unit, Center for Prevention, Diagnosis and Treatment of Infertility, Careggi University Hospital, 50139 Florence, Italy; 2Department of Experimental and Clinical Biomedical Sciences “Mario Serio”, University of Florence, 50134 Florence, Italy; 3Department of Experimental and Clinical Medicine, University of Florence, 50134 Florence, Italy

**Keywords:** sperm cryopreservation, fertility preservation, sperm DNA damage, assisted reproduction, gonadotoxic treatments, cryoprotectants

## Abstract

Cryopreservation is an expanding strategy to allow not only fertility preservation for individuals who need such procedures because of gonadotoxic treatments, active duty in dangerous occupations or social reasons and gamete donation for couples where conception is denied, but also for animal breeding and preservation of endangered animal species. Despite the improvement in semen cryopreservation techniques and the worldwide expansion of semen banks, damage to spermatozoa and the consequent impairment of its functions still remain unsolved problems, conditioning the choice of the technique in assisted reproduction procedures. Although many studies have attempted to find solutions to limit sperm damage following cryopreservation and identify possible markers of damage susceptibility, active research in this field is still required in order to optimize the process. Here, we review the available evidence regarding structural, molecular and functional damage occurring in cryopreserved human spermatozoa and the possible strategies to prevent it and optimize the procedures. Finally, we review the results on assisted reproduction technique (ARTs) outcomes following the use of cryopreserved spermatozoa.

## 1. Introduction

The possibility to cryopreserve gametes and embryos represents an important advancement in reproductive biology. Such procedures are indeed essential for the maintenance of endangered animal species, for animal breeding via artificial insemination and, importantly, to give hope for future parenthood to individuals who must undergo therapies or surgery which can compromise gonadal function. In particular, according to the last edition of the WHO laboratory manual for the examination and processing of human semen [1], fertility preservation should be offered for autologous use to men before treatments with cytotoxic agents or radiotherapy [2], vasectomy, social freezing in cases of active duty in a dangerous occupation, male-to-female transsexual adults and adolescents before the initiation of hormonal therapies. In addition, semen cryopreservation can be advised to men before assisted reproduction techniques (ARTs) in the case of patients being unable to ejaculate, or with severe oligozoospermia or the inability to provide a fresh sample on the day of the ART procedure. Finally, the technique is used to cryopreserve spermatozoa from healthy donors for future use in couples where the male partner is azoospermic, to prevent the transmission of an inherited disorder, for women who wish to conceive but do not have a partner (the latter in those countries where the procedure is allowed) or for lesbian and transgender couples. Figure 1 reports the various conditions where cryopreservation is advised.

From the first attempts to cryopreserve human male gametes, many advancements have been made, effective cryoprotectants have been discovered, the possibility to cryopreserve in liquid nitrogen has been developed and, nowadays, semen cryobanks are distributed widely around the world.

There are several procedures/protocols to cryopreserve semen and spermatozoa in liquid nitrogen or vapors (for review, see [3] and Figure 2). The research in this field has been focusing on finding solutions to minimize the generation of ice crystals within the cytoplasm, leading to the development and use of two types of cryoprotectants, permeating and non-permeating. The former (including DMSO, glycerol, ethylene glycol and others) creates an osmotic gradient to limit the formation of ice and stabilize the lipid bilayer. Non-permeating cryoprotectants (including sugars and lipoproteins) contribute to water leakage from the cytoplasm and protect membrane integrity. Cryoprotectants used nowadays usually include glycerol, a sugar and egg yolk mix used as a non-permeating cryoprotectant [4]. Antibiotics are also added to the mixture to fight the detrimental effect of microorganisms that may be present in semen.

Another critical point in the process of cryopreservation is represented by the cooling rate, which should be controlled, for instance by the use of programmable freezers or through a standardized manual fast vapor freezing method [1]. Similarly, the thawing process is also critical, and different thawing methods can be used. At present, the WHO manual advises to proceed with fast thawing at 37 °C [1]. It should also be noted that semen processing before cryopreservation should be performed in a sterile environment [1] according to good manufacturing practice (GMP) guidelines, to avoid additional contaminations to those already present in semen that may further compromise sperm performances.

Based on their particular structural and morphological characteristics, including the low amount of cytoplasm, spermatozoa are considered to be quite resistant to potential cryodamage [5]. Due to the particular lipid composition of the plasma membrane, which contains higher levels of cholesterol and a lower ratio of unsaturated vs. saturated fatty acids, human spermatozoa are less susceptible to cryodamage with respect to the spermatozoa of other mammals [6]. Despite this, human spermatozoa may be heavily damaged by the freezing/thawing procedure both at structural and functional levels (see below). There is evidence that the susceptibility to the deteriorating effects not only depends on the initial quality of semen [7] but, also, may differ from one subject to another and vary among the different types of pathology for which cryopreservation is indicated [7,8,9,10]. At present, the intrinsic sperm characteristics responsible for the different susceptibility to cryodamage are not known. Furthermore, the issues producing the damage should be better defined. Besides intracytoplasmic ice formation, the generation of reactive oxygen species (ROS) is considered to be one of the main causes responsible for the damage [11] but whether other toxic products are generated during the procedure is less known and poorly studied.

Although most studies agree that sperm damage is induced by freezing and thawing processes per se rather than by long storage in liquid nitrogen [12,13,14], there is at least one study reporting storage–time-dependent structural damages [15]. Importantly, the attainment of pregnancy with long-term cryopreserved spermatozoa has been reported [16,17] and a recent study demonstrated that storage up to 15 years does not affect clinical ART outcomes with donor spermatozoa [18].

Semen banks must use safety procedures in order to prevent infectious disease transmission with cryopreserved semen and the risks of cross-contamination inside storage tanks. For this reason, men should be screened for the main transmitted viral diseases (Hepatitis B or C, CMV and HIV) and other pathogens according to local legislations. For virus/pathogen positive samples, it is advised to use separate tanks and other strategies to avoid cross-contamination [1]. The recent SARS-CoV-2 pandemic raised some questions regarding semen cryopreservation safety for COVID-19-affected individuals [19,20]. However, the occurrence of SARS-Cov2 mRNA in human semen has been only occasionally reported [21,22], and whether the virus may be transmitted through semen remains to be defined. A recent study [23] reporting the results of a survey administered to 22 European semen banks showed that the majority of them did not adopt particular safety measures during the pandemic period and the most common strategy consisted of the administration of an anamnestic questionnaire to patients, as only half of the centers required a nasopharyngeal swab.

The present review describes the structural, functional and molecular damage of cryopreserved human spermatozoa with a focus on “omics” studies. In addition, we review recent studies with new strategies to prevent or limit the damage as well as novel approaches to sperm preservation. Finally, a paragraph is dedicated to the outcomes of ARTs using cryopreserved semen.

## 2. Functional and Structural Damage

Several damages may occur to the sperm structure during freezing and thawing processes. Despite some controversy in the literature [24], alterations in sperm morphology following cryopreservation with various protocols have been reported in several studies. Both the percentage of abnormal spermatozoa [25,26,27] and the teratozoospermia index (TZI, i.e., the number of abnormalities/spermatozoon) may increase after the procedure [28]. Most alterations involve tail and acrosome structures.

Considering that the acrosome is of fundamental importance during the process of fertilization both in vivo and in vitro [29,30], alterations in acrosomal structure, the depletion of its content and a loss of the ability to respond to stimuli may severely compromise sperm functionality. Several studies have reported ultrastructural acrosome alterations, including wrinkling of the plasmalemma, loss of acrosomal content and vesiculations [31], likely due to ice crystal formation, osmotic changes, ROS generation and an influx of liquid due to membrane alterations occurring during the process. Gomez-Torres et al. [32] reported an increase in spontaneous acrosome reaction in a high percentage of spermatozoa both for high- and low-quality semen samples using a test yolk/glycerol buffer as an extender and a slow freezing protocol. An increase in the spontaneous acrosome reaction may impair the fertilizing ability when cryopreserved spermatozoa are used in more physiological assisted reproduction procedures such as first level intrauterine insemination (IUI) or second level in vitro fertilization (IVF), where, different from intracytoplasmic sperm injection (ICSI), the integrity of the acrosome is required to allow the penetration of oocyte vestments. In a subsequent study [28], the same group reported that the employment of a vitrification protocol strongly reduces the spontaneous post-thawing acrosome reaction.

Tail morphological alterations might be responsible for the decrease in sperm motility observed after thawing and may occur in a high percentage of spermatozoa [28]. Such alterations are mostly due to osmotic changes during the process (leading to the coiling up of the tail), or an incorrect distribution/content of tubulins [15,32,33], which are the major component of the microtubules.

Sperm motility decrease can also be due to altered mitochondrial structure and function that may be severely compromised by cryopreservation. Mitochondria play important roles in spermatozoa, being involved in the generation of ATP to support motility and other functions requiring energy [34]. Electron microscopy studies have demonstrated several post-thawing alterations, including loose structures, widened crests and the appearance of vacuoles [35]. The mitochondrial membrane may be also damaged and its fluidity may be severely modified. All of these alterations may be responsible for the decreased mitochondrial membrane potential (MMP) observed in various studies [36,37], as well as in the generation and release of ROS by the organelle [38].

Cryopreservation severely damages sperm motility and viability. The decrease in the percentage of motile spermatozoa varies from 30 to 50% depending on the freezing/thawing protocols as well as on the initial semen quality [7,39,40]. In particular, our study [7], conducted on 788 semen samples from patients cryopreserving for testicular and hematological cancers, oligozoospermia and other pathologies demonstrated that initial semen quality predicts with high accuracy, sensitivity and specificity motility recovery after thawing. Moreover, we showed that total and progressive motility recovery was, on average, close to 0% for samples where at least one parameter (motility, concentration or morphology) of the initial semen quality was below the fifth percentile of WHO 2010 reference limits [41]. In such a situation, finding a motile spermatozoon may become a hard task for ICSI operators. In particular, the situation is critical for patients cryopreserving for testicular cancer, whose basal semen quality may already be compromised [42,43,44,45]. It is well known that when ICSI is performed with an immotile spermatozoon, where viability cannot be guaranteed, the outcomes may be worse [46,47]. Similarly to motility, sperm viability also decreases following cryopreservation in a variable measure (from 30 to 50 depending on the study) [7,48,49,50,51]. There is evidence that ROS production during the cryopreservation procedure may induce an apoptotic pathway [52,53], but other issues leading to sperm death cannot be excluded such as the peroxidation of fatty acids in the plasma membrane [53,54,55].

The occurrence of structural and functional alterations following cryopreservation may condition the choice of the technique in ART laboratories, such as the different procedures requiring a minimum number of motile spermatozoa. For instance, for IUI, a minimum of 1.5 × 10^6^ progressive motile spermatozoa are required, as there is evidence of lower outcomes below such a number [56,57]. The minimum requirement of progressively motile spermatozoa for the IVF procedure is lower (between 50 and 100 thousands/oocyte), but even such a procedure can be hampered in the case of cryopreserved spermatozoa, leading to the forced choice of ICSI application. In these cases, where no motile spermatozoa are found after thawing, chemical inducers, such as pentoxyfilline [58], dimethylxantines, myoinositol and mechanical or laser-assisted maneuvers [59], can be used to induce tail movement in viable immotile spermatozoa to select them for oocyte insemination. Such procedures have been successfully used in IVF laboratories producing live embryos [59].

In cases of low semen quality, the bank should assure an adequate number of cryopreserved devices [1] by banking repeated ejaculates and appropriate counseling should be given to the patient regarding the possibilities for future use in ARTs.

## 3. Damage to DNA

The integrity of sperm DNA plays a major role in offspring health and development [60]. Among the different DNA anomalies that can be present in the male gamete, DNA fragmentation is the most frequent [60]. Several studies have demonstrated that men with a high percentage of sperm DNA fragmentation (sDF) have less chance of success both in in vivo [61,62,63,64] and in vitro reproduction [65,66,67]. This type of damage may be further increased by in vitro manipulation [68] and, in particular, the negative impact of sperm cryopreservation on sDF is well documented [69,70,71,72,73,74,75], although there is not a general consensus. Indeed, a few studies did not find significant differences in sDF levels between fresh and cryopreserved semen samples [76,77]. Such discrepancies may depend on the technique used to evaluate sDF [78].

The extent of the damage is highly dependent on the individual cryotolerability. Sperm DNA from infertile men is more susceptible to freezing damage than that from fertile men [79]. Cancer patients, those who most of all resort to cryopreservation, show higher average sDF values with respect to fertile or healthy age-matched men [80,81,82,83]. In these cases, additional damage can be introduced with cryopreservation, making the topic of fertility preservation even more problematic.

Different mechanisms have been suggested as the cause of the increase in sDF after cryopreservation. Some studies have reported an increase in caspase activity during the cryopreservation process, leading to the hypothesis of an involvement of apoptotic pathways in sperm DNA cryo-injury [77,84]. However, adding caspase inhibitors to cryoprotectant medium, Thomson and colleagues [69] did not observe a reduction in the amount of DNA damage generated during cryopreservation, suggesting that caspases are probably not responsible for the observed sDF increase. As mentioned above, during sperm cryopreservation and thawing, increased ROS production [85] and decreased antioxidant levels [86] are known to occur. This imbalance in the redox state could be responsible for most of the deleterious effects on DNA observed after these procedures. Changes in mitochondrial membrane fluidity may trigger an excessive release of ROS and a consequent increase in oxidative stress [87]. For this reason, different mitochondria-targeted antioxidants have been added to freezing media in order to improve post-thaw sperm motility and decrease ROS levels. Among these, Elamipretide, Vitamine E and L-carnitine [39,88,89] have been shown to have some efficacy in protecting sperm DNA from cryodamage.

An excess of ROS can induce DNA base modifications. 8-hydroxy-2′-deoxyguanosine (8-OHdG) is one of the more abundant forms of free-radical-induced oxidative lesions, being widely used as a biomarker of oxidative DNA damage. A significant increase in the percentage of 8-OHdG-positive spermatozoa was observed after cryopreservation [69,72]. ROS-induced DNA base modifications may facilitate genome instability and mutations by limiting the DNA repairing capacity of spermatozoa [90]. Studies on other cell types have demonstrated that beyond the mutagenic effect, oxidized DNA bases can negatively affect the methylation of adjacent cytosines [91,92]. Excessive ROS production, on the other hand, may up-regulate DNA methyltransferases (DNMT), triggering hypermethylation [93]. It is conceivable that ROS may also alter the sperm epigenetic status. More comprehensive studies on non-human mammalian species, focusing on the global epigenome, support this hypothesis, demonstrating significant changes in sperm DNA methylation after cryopreservation [94,95,96,97]. These data give rise to concerns, considering that sperm cryopreservation is frequently used in ART and that sperm DNA methylation may play a decisive role in offspring health [98]. Results on human spermatozoa published so far seem more reassuring because they have not shown significant alterations in the DNA methylation pattern after cryopreservation [99,100,101]. In particular, Khosravizadeh investigated the effects of cryopreservation on DNA methylation in specific genes in the chromosome 15q11–q13 region. It is known that epigenetic changes in this region lead to imprinting disorders including Angelman syndrome (AS) and Prader–Willi syndrome (PWS), two distinct neurogenetic disorders [102,103]. Although an increase in intracellular ROS levels and sDF in cryopreserved compared to fresh spermatozoa was observed, DNA methylation of the selected gene regions was not affected. A recent study, through an analysis of alternative splicing, concluded that the cryopreservation of human spermatozoa can cause epigenetic instability in a small percentage of patients [104]. Clearly, more studies with a larger sample size and different cryopreservation protocols are needed to resolve the issue of whether the sperm epigenetic pattern is altered by cryopreservation.

Freezing–thawing may also impact the degree of sperm chromatin condensation. Sperm chromatin decondensation at the right time is essential for the zygote formation and reduced condensation may predispose sperm DNA to oxidative damage. Royere and colleagues observed abnormal ‘overcondensation’ of the chromatin after cryopreservation via acridine orange staining and Feulgen DNA cytophotometric studies [105,106]. Other studies, however, revealed an increase in sperm nuclear chromatin decondensation after cryostorage, both through high-magnification microscopy analysis [107] and aniline blue staining [108]. Similar results were obtained by Rarani and colleagues [109] by using aniline blue and toluidine blue staining in semen samples from 30 normozoospermic men. Finally, a study demonstrated that not only the cryopreservation process alters chromatin compaction, but that this damage is also dependent on the storage time [110], as it was found that this further increased after 90 days of storage. Overall, most studies seem to indicate that cryopreservation procedure may decondense sperm chromatin, likely contributing to increased oxidative damage to DNA.

Cryopreservation can also induce genetic damage. Valcarce and colleagues [111] studied the effect of sperm cryopreservation on six genes with roles in fertilization and embryo development (BIK, FSHB, PRM1, ADD1, ARNT and PEG1/MEST) and two genome regions related to Prader–Willi and Angelman syndrome (UBE3A and SNORD116/PWSAS). The results of this study showed a detrimental effect in some important imprinted regions of these genes, highlighting a different vulnerability of the genome to the damage; the maximum number of lesions was detected in SNORD116/PWSAS, PRM1, UBE3A and BIK, whereas the minimum number of lesions was detected in PEG1/MEST and FSHB [111].

## 4. Damage to mRNAs, Proteins and Metabolites

As described, cryopreservation provokes both structural and biochemical damages to spermatozoa which may lead to an impairment of fertilization ability or embryo development [52].

Recent advances in omics coupled with bioinformatic approaches could be useful to uncover molecular and cellular changes in sperm molecules, such as RNAs and proteins. Discovering new possible markers of sperm tolerance to freezing/thawing could be helpful to optimize the current cryopreservation protocols. Most studies investigating these new frontiers were performed in animal models such as bovine and pig above all because in these animals artificial insemination mainly uses frozen spermatozoa.

Although mature spermatozoa are considered to be transcriptionally silent, recent evidence has demonstrated the presence of various types of RNAs representing epigenetic marks involved in spermatogenesis but also transmitted to the next generations, reflecting paternal environmental exposure [112].

Transcriptome analysis using RNA sequencing showed a differential expression of mRNAs and miRNAs between fresh and frozen–thawed boar and bull spermatozoa [113,114] suggesting that this procedure could alter the abundance of sperm transcripts related to fertility-associated functions. In humans, little is known about sperm RNAs after cryopreservation. A recent study of the group of Isachenko revealed differentially expressed genes comparing fresh, conventionally frozen and vitrified spermatozoa [115]; however, the two cryopreservation methods seem to be epigenetically safe, although vitrification induced minor changes in terms of fertilization impact with respect to the conventional procedure. Valcarce and colleagues observed a significant reduction in most analyzed mRNAs chosen among the considered markers of male quality and pregnancy success, hypothesizing that cryopreservation can commission transcripts to degrade [116].

Among sperm mRNA, miRNAs and small non-coding RNA not only regulate post-transcriptional gene expression and mediate epigenetic inheritance in spermatozoa but also contribute to maternal mRNA degradation after fertilization [117]. Moreover, different studies have demonstrated the role of miRNA in mammals’ embryo development [117,118]. Only two studies investigated the expression of miRNAs after cryopreservation in human spermatozoa, finding a downregulation of those implicated in fertilization [119,120].

The advent of proteomic analysis has allowed the evaluation of the protein profile and its changes in certain physiological or pathological contexts. To date, only a few studies have evaluated the effects of cryopreservation on the human sperm proteome [121,122].

The first study on the proteomic profile of cryopreserved human spermatozoa from sperm donors published by Wang and colleagues [122] identified twenty-seven differentially expressed proteins mostly associated with sperm motility, acrosome integrity, capacitation, viability and mitochondrial activity [122].

Subsequent studies, using high-throughput technologies, demonstrated a higher number of differentially expressed proteins [121,123]. Moreover, in these studies, a change in sperm proteomic profile has been reported not only by comparing fresh and frozen spermatozoa but also by evaluating different methods, steps and actors involved in the cryopreservation process [121,124]. Differentially expressed proteins have been found by examining both different preservation methods, such as slow or rapid freezing and vitrification [125,126], and by modifying freezing/thawing conditions [121], as well as by using different types of cryoprotectants [121] or different cryopreservation carriers (cryovials or cryostraws [124]). Although the identity of most of the up- or down-regulated proteins found in the different experimental conditions varies in these studies, they are almost always involved in important sperm functions for achieving oocyte fertilization, such as energy production, motility, apoptosis, DNA damage and repair, capacitation, acrosome reaction, sperm–egg fusion, etc. Some of these proteins could be revealed as potential markers of sperm freezability once validation studies are performed. An example is represented by the Arylsulfatase A, a protein implicated in gametes fusion, whose levels decreased after both slow- and rapid-freezing, suggesting that it could be used as a marker of sperm quality after cryopreservation [126,127]. Of note, it has been reported that oncological patients show already altered proteomic profiles before cancer treatments [128,129,130] and, therefore, at the moment of cryopreservation. In such a situation, eventual cryodamage to protein expression is added to already-existing damage.

It should be highlighted that the heterogeneity in proteins found among the studies could also be ascribed, besides the different conditions of cryopreservation procedure (type of cryoprotectant, freezing method, thawing procedure and temperature, etc.), to the protein analysis procedure as well as the use of whole semen or selected spermatozoa. Indeed, semen is a heterogeneous fluid also containing non-sperm cells that can contribute to the proteome modifications.

Recent advances in mass-spectrometry have allowed for also exploring, besides the cell transcriptome and proteome, the metabolome, offering the opportunity to assess and quantify small molecules as the final results of transcriptional and translational events. This approach can facilitate the understanding of the mechanisms of biological and biochemical processes, answering the specific research questions. A few studies, most of which were conducted on animal species, have investigated the differences in metabolomic profiles between fresh and frozen spermatozoa, finding the significant deregulated metabolites implicated in the main sperm energy pathways [131,132,133]. Interestingly, the only study on human spermatozoa identified 16 significantly deregulated metabolites between fresh and post-thawed spermatozoa, suggesting that metabolomic changes during the cryopreservation process could be helpful in order to identify new markers of human sperm freezability [123].

## 5. Possible Strategies to Prevent the Damage and New Approaches to Sperm Cryopreservation

As mentioned, although cryopreservation is a valuable option for preserving male fertility when required, cryo-injury represents a problem for the future use of cryopreserved gametes, especially when basal semen characteristics do not guarantee the effectiveness of the procedure. In such situations, semen banks should adopt strategies aimed to prevent or to reduce the cryodamage.

One possible strategy is represented by the pre-cryopreservation selection of motile spermatozoa with standard procedures such as density gradient centrifugation (DGC) or swim up, which eliminate dead, immotile and morphologically abnormal spermatozoa as well as immature germ cells and leucocytes that can be present in whole semen. Such a strategy can be adopted in particular situations (for example, when high levels of leukocytes are present [1]) or in the case of elevated percentages of immotile spermatozoa. Leucocytes, apoptotic/damaged spermatozoa and immature germ cells may indeed produce high levels of ROS, aggravating the damage induced by cryopreservation. Alternatively, and when possible, the IVF laboratory may attempt post-thawing selection in order to enrich the sample with motile spermatozoa. It should be considered that selection procedures such as DGC and, to a lesser extent, swim up, may induce damage to DNA per se [134,135] and that removal of seminal fluid eliminates the protective effects of antioxidant substances present in semen [136]. As a matter of fact, an increase in post-thawing DNA damage has also been reported when swim up or DGC procedures were used to select spermatozoa before cryopreservation, and no improvement in sperm motility was observed [70]. However, a recent study demonstrated that performing DGC selection before cryopreservation resulted in better post-thaw parameters with respect to selection after thawing [137].

Improvements in sperm parameters, including decreased DNA damage, have been reported when DGC was followed by a more sophisticated sperm selection procedure such as annexin V-magnetic assisted cell sorting (MACS) both pre- or post-cryopreservation [138,139]. In particular, post-thawing selection procedures have been attempted in a few studies, reporting an improvement in sperm motility [140,141]. Successful live births after sperm sorting with annexin V-MACS of cryopreserved spermatozoa with high levels of sDF from a cancer patient survivor were reported [142]. The paucity of studies on pre- or post-cryopreservation sperm selection, however, does not allow one to draw firm conclusions regarding whether they can be applied on a large scale.

In view of the fact that ROS generation during the freezing/thawing process is the main thing responsible for cryodamage, several studies have evaluated the effects of the addition of natural agents with antioxidant properties (vitamins, endogenous substances, herbal extracts, antioxidant enzymes and others) to semen extenders (reviewed in [4,143]) with the aim of mitigating the possible toxic effects of extenders. Most of these studies reported some efficacy in sperm parameters and DNA integrity, but no clear-cut conclusions could be drawn and the need for further studies was evidenced [4,144]. Kumar et al. [125] have shown that mitoquinone, a mitochondrial-targeted antioxidant, both attenuates ultrastructural changes and protects several proteins involved in sperm key functions from alterations induced by vitrification. Emerging studies in the last few years have investigated the effects of the supplementation of freezing media with taurine and hypotaurine [145,146], melatonin [147,148] and gallic acid [25] as antioxidants. Some beneficial effects have been reported with taurine and its precursor hypotaurine which slightly but significantly improved sperm parameters, including DNA integrity, when supplemented to extenders both for standard cryopreservation [146] and vitrification [145] methods. Among the tested antioxidant agents, the most efficient in mitigating the cryodamage in human spermatozoa was melatonin, as reported in a recent meta-analysis [144]. Indeed, the addition of the hormone, a physiological regulator of the circadian rhythm, to cryoprotectants exerts a significant positive effect on sperm progressive motility and viability [144], is currently used for the cryopreservation of spermatozoa from several animal species [149,150,151] and has been also shown to improve the survivability of oocytes and embryos [148]. Such beneficial effects are not surprising considering that melatonin, besides showing antioxidant activity, is an anti-apoptotic and ROS scavenging agent [147].

A recent study has demonstrated that preconditioning sperm cells before cryopreservation with sublethal nitric oxide levels not only improves sperm motility, viability and fertilizing capability [152] but also maintains the redox balance without altering the metabolism of sperm proteins [127].

Another approach recently investigated concerns surrounding the addition of growth factor- or platelet-rich plasma to cryoprotectants. Mirzaei et al. [51] demonstrated that the addition of a plasma rich in growth factors at different percentages (from 1 to 10%) could significantly improve sperm parameters and DNA integrity with the best results at 1% concentration. The authors attribute such positive effects to the action exerted by growth factors on their receptors on the human sperm surface more than a direct antioxidant or ROS scavenging effect [51]. Minimal improving effects were observed with platelet-rich plasma [49,153] in a small number of samples. No studies so far have addressed the question of whether cryopreserved spermatozoa with enriched plasma retain the ability to fertilize and support embryo development.

Vitrification has been successfully used to cryopreserve oocytes and embryos [154]. The group of Isachenko first introduced this method [5,155,156,157] which is based on the direct exposure of the sample to liquid nitrogen, allowing for ultrarapid freezing that avoids or strongly reduces the formation of ice crystals in the cell. In the case of spermatozoa, both the direct plunge of semen (after dilution with an extender) in liquid nitrogen or after aspiration in closed devices (in straw vitrification) can be performed. Vitrification is easy to perform, is less time-consuming with respect to the standard procedure and can be applied both for whole semen and selected spermatozoa free of seminal plasma. Clinical studies have demonstrated that vitrified spermatozoa retains its fertilizing ability both in IVF, ICSI and IUI techniques, achieving live births [158]. A recent meta-analysis evidenced some advantages in post-thawing parameters after vitrification with respect to conventional methods [159]. In particular, progressive motility and morphology appear to be better preserved. Concerning DNA damage, although some studies promote vitrification [160], other authors have not observed differences in post-thaw DNA damage between the two methods [161,162,163,164]. The heterogeneity of studies does not allow one to draw firm conclusions on whether vitrification should be preferred to the standard cryopreservation [163,164,165,166] and, at present, vitrification for human spermatozoa should be considered to be experimental [1]. A recent study analyzed post-thawing parameters after vitrification vs. vapor fast freezing of low semen volumes in different experimental conditions including the use of cell sleepers [167] (also see below). They showed that vapor fast freezing better prevents cryodamage independently of the type of cryoprotectant and the support used. It should be noted that vitrification can present some disadvantages with respect to conventional cryopreservation, such as the higher concentration of cryoprotectants used (increasing their toxicity), a higher risk of potential contamination with pathogens (requiring sterilization of liquid nitrogen) and, finally, requiring skilled operators for manipulation procedures [168].

The attainment of the successful generation of embryos [169] and even live births [170,171,172] in some mammalian species after sperm lyophilization (freeze-drying) is certainly attractive for semen banks and ART centers. Lyophilization is indeed a more sustainable technique which would avoid the use of expensive liquid nitrogen, allowing easy storage, packaging and transfer of the samples. At present, only few studies, with conflicting results, have evaluated the eventual damaging effect of lyophilization on human spermatozoa. Kusakabe et al. [173] demonstrated that only a low percentage of sperm showed chromosomal alterations and Gianaroli et al. [174] did not find increased DNA damage after dry storage with respect to the standard procedure. However, lyophilization may harm cell membranes [175] and produce detrimental effects on the sperm head [176]. Considering that after lyophilization spermatozoa do not preserve viability or motility, the fact that they can support embryo development and live births after ICSI in some mammalian species (see above) indicates that the maintenance of DNA integrity [173,174] is an important achievement of the freeze-dry procedure. Whether non-viable spermatozoa may support embryo development and live birth in humans as well is presently poorly known, as only a case report on the attainment of live birth with an unviable testicular spermatozoon [177] is present in the literature. Clearly, further studies are needed regarding this interesting and sustainable method of sperm storage, which, if successful, could open important perspectives both for human and animal reproduction.

In view of the variety of studies regarding the additional components to be added to standard cryoprotectants and different procedures to freeze/thaw spermatozoa, it is not possible at present to define the optimal mixture of cryoprotectants and the best freezing procedure. Hopefully, further well-designed comparative studies or metanalyses will help to define a gold standard procedure for semen or sperm cryopreservation.

Finally, it is worth mentioning that in the last edition of the WHO laboratory manual for examination and processing of human semen [1] it is stated that “as only a single spermatozoon is needed for ICSI of each oocyte, cryopreservation of any live spermatozoon is worthwhile”. The cryopreservation of small sperm numbers can be of clinical value for some male infertility factors such as severe oligozoosermia or criptozoospermia, cryptorchidism and obstructive azoospermia. Clearly, the use of standard procedures for very low sperm numbers is inadequate and may be time-consuming for the ICSI operators due to the dilution with the cryoprotectant, but also considering the cryodamage (see above). There are some alternative strategies to cryopreserve low sperm numbers, including the use of biological or non-biological carriers [178]. In particular, the latter (cryoloops, cell sleepers, cryotops and others) appear to be quite promising as they allow for the recovery of good percentages of motility and viability [167] and can also be used for spermatozoa recovered after TESE [179]. Such methods have been used in clinical settings demonstrating their efficiency in supporting live birth [180,181,182]. One important drawback of cryopreserving single spermatozoa is the necessity of using a micromanipulator with ICSI needles to pick single spermatozoa, requiring skilled operators, time and expensive instrumentation.

All of the possible strategies to prevent cryopreservation-induced damage and new alternative approaches are represented in Figure 2.

## 6. ART Outcomes after Use of Cryopreserved Spermatozoa

The usage rate of cryopreserved spermatozoa after cancer survival is quite low, estimated between 3 and 10% depending on the study and the length of follow up [42,183,184,185,186]. Such a rate is even lower for patients cryopreserving in a prevision of an ART procedure because, if present, fresh semen is always preferred. In most cancer cases, cryopreserved semen is destroyed after patient death, the attainment of a natural pregnancy or because of restored fertility after chemo- or radiotherapies. Regarding the latter point, studies on juvenile hematological cancers [187,188] indicate that most Hodgkin and non-Hodgkin lymphoma patients recover spermatogenesis 2 years after therapies, although the recovery highly depends on the therapy regimens (with worse results when chemotherapy is associated with radiotherapy) and is unpredictable. Similar results have been reported for testicular cancer patients [189]. One important aspect is related to the possible effects of chemo- and radiotherapies on sperm DNA integrity, also considering that there are studies reporting higher sperm DNA fragmentation levels in cancer patients before any therapy (see above). Most studies report an increase in sDF post-chemo or radiotherapies in testicular (reviewed in [190]) and hematological [80,81] cancer patients, which may persist for years after the end of the therapies. In consideration of the fact that cryopreservation may damage sperm DNA per se (see above), important clinical questions arise about the opportunity to use cryopreserved or fresh semen in cases of the recovery of spermatogenesis and when it is the right moment to attempt natural conception after cancer treatments, in order to avoid/decrease the risk of transmitting defective paternal genome to the offspring. Regarding the second question, as mentioned above, couples whose male partner regains fertility are requested to wait 1–2 years after the last cycle of therapy before attempting to conceive naturally or by ART using fresh semen. In any case, larger follow up studies are requested to give precise answers. Regarding the first question, it should be considered that the results of studies evaluating ART outcomes and the health of offspring with cryopreserved semen from cancer patients are highly conditioned and limited by the low usage rate of cryopreserved semen. In the bulk of them, these studies are quite reassuring about the attainment of clinical pregnancy and healthy offspring, with rates that do not differ or are only slightly lower with respect to control cycles [45,190,191,192].

Few studies have compared ART outcomes with fresh and frozen semen. In a randomized prospective study, Kuczynski et al. [193] demonstrated that the use of frozen spermatozoa from men with poor semen quality in ICSI cycles resulted in similar outcomes to freshly ejaculated spermatozoa and, actually, the rate of ongoing pregnancies was slightly, although insignificantly, higher in the frozen group. Similarly, a recent systematic review [194] on the use of fresh or frozen testicular spermatozoa from non-obstructive azoospermic men did not evidence significant differences in fertilization or pregnancy rates after ICSI. It should be noted, however, that Hauser et al. [195] reported, on average, lower implantation rates with frozen testicular spermatozoa. A retrospective study by Zhu et al. [196] compared the results of a consistent number of cycles from the fresh semen of normozoospermic men to those obtained with donor frozen semen. Clinical pregnancy and live birth rates after IVF were significantly higher and birth defects were reduced in the donor group.

Lower outcomes in terms of pregnancy rates are achieved when intrauterine insemination (IUI) is used in ART cycles. Botchan et al. [184], comparing the outcomes of ICSI and IUI cycles with the frozen spermatozoa of 184 cancer patients, found significantly higher pregnancy rates in the former (37.4 vs. 11.5%). Pregnancy rates per IUI cycle are also lower when cryopreserved donor semen samples are used [197,198,199]. Overall, these results suggest that IUI should be employed only in cases of attainment of adequate semen quality after thawing [184].

## 7. Conclusions

In this review, we highlight the various aspects concerning the sperm cryopreservation process, including the categories of subjects to whom it is recommended, the methods currently used in clinical practice (summarized in Figure 1) and also the experimental or promising new procedures that could be used in order to reduce cryodamage and to improve the efficiency of the process (Figure 2).

Although several advances in the study of cryopreservation since its discovery have been made, further research is still needed to improve all of the critical points. The knowledge acquired so far on sperm structural and functional damages as well as on alterations of sperm DNA and chromatin integrity should be used as the foundations for the optimization of the protocols. In particular, it will be necessary to perform comparative studies to define which are the most favorable freezing method(s), the optimal cooling and thawing conditions, the best cryoprotectant(s) and whether supplements of antioxidants or other substances to extenders can be of help to minimizing cryodamage. Recent results on melatonin and platelet-rich plasma appear to be quite promising but further studies are needed.

The characterization of genes, transcripts, proteins and metabolites that change their expression following cryopreservation could identify potential markers of sperm susceptibility to cryoinjury. Such markers could be used to identify samples at risk of higher cryodamage allowing for better patient counseling, and could be used as targets for the development of possible additions to cryoprotectants to prevent the damage. In addition, omics studies conducted until now evidence that every single step of the cryopreservation process should be carefully analyzed in order to understand which are the critical ones for the generation of ultrastructural alterations as well as epigenetic, transcriptomic and proteomic cryoinjuries. Therefore, each step is part of a puzzle where every single piece contributes to determining the future of each spermatozoon after freezing and thawing. We cannot exclude that the critical steps may vary in relation to different semen quality or categories of patients. Ideally, future research should be devoted to identifying “personalized” solutions. Of note, one limitation of “omics” studies in the case of semen is the concomitant presence of non-sperm cells which can condition the results. So far, only few studies have attempted to eliminate non-sperm cells from the analysis.

It must be noteworthy that a cryopreserved male gamete with good motility and morphology that is usually chosen to fertilize the oocyte via the ICSI procedure could hide injuries not identified with semen analysis, impair fertilization and/or subsequent embryo development and even damage the progeny. Thus, if so far only sperm motility and viability are usually evaluated after cryopreservation, the occurrence of “hindered” damages makes it necessary to introduce second level analyses to understand further aspects before ART application. Finally, although available studies are quite reassuring regarding the use of cryopreserved spermatozoa in ART, reporting no or only slightly lower outcomes with respect to control cycles, they are heterogeneous and with limited caseloads. Larger, multicenter studies with extended follow ups of offspring are necessary in order to better understand whether cryopreserved spermatozoa have the same outcomes as fresh ones.

## Figures and Tables

**Figure 1 ijms-24-04656-f001:**
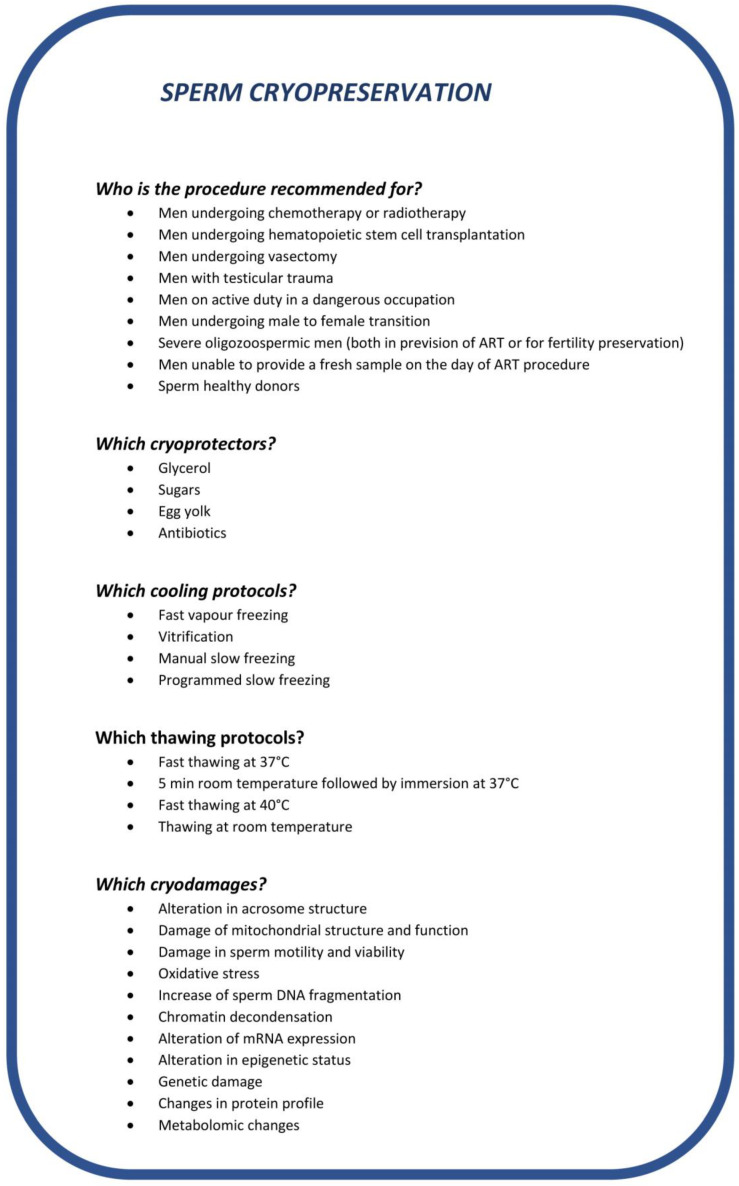
Schematic representation of categories of subjects to whom sperm cryopreservation is advised, cryoprotectants, cooling and thawing protocols which can be used and cryodamages that may occur.

**Figure 2 ijms-24-04656-f002:**
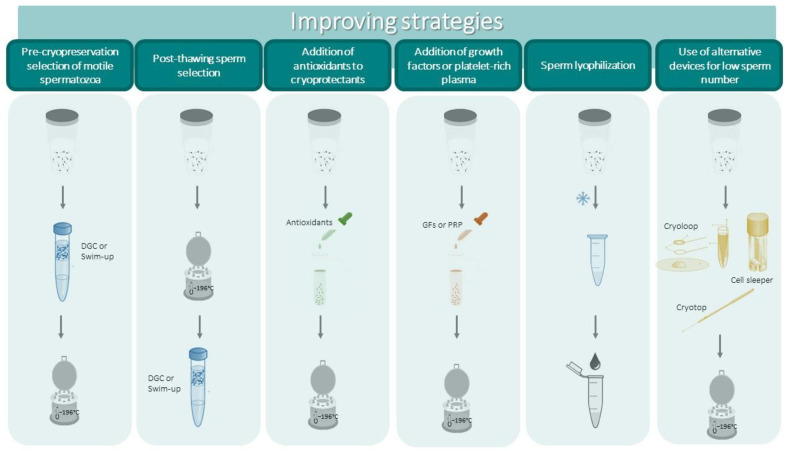
Schematic representation of strategies to improve the outcomes of cryopreservation and to cryopreserve low sperm numbers.

## Data Availability

Not applicable.

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
