# Peer review of "Cryopreservation of Human Spermatozoa: Functional, Molecular and Clinical Aspects"

_ijms, 2023, doi:10.3390/ijms24054656_

Round 1

Reviewer 1 Report

The authors present an up to date review of structural, functional and molecular changes to sperm as a result of cryopreservation, potential solutions and impact on human ART outcomes. Some of the information presented is very basic and could be trimmed from the review to leave more room for discussion of recent developments. Overall, the review is of good quality and my suggestions for the authors to consider are below:

Paragraph structure needs improvement – some very long (55-82) and very short (107) paragraphs

Unclear why reference to animal conservation and agricultural breeding is mentioned if the focus is human ARTs

L42-43 lesbian and transgender couples are also common users of donor semen

L52 “cryoprotector” is not a commonly used term, suggest replacing with “cryoprotectant” throughout

L56-57 reference?

L63 reference?

L66 reference?

L69 antibiotics are part of cryomedia but aren’t a cryoprotectant

L71-75 references?

L50-82 presents a lot of very basic information on semen cryopreservation which is present in many other reviews and not really necessary

L108-116 relevance of this paragraph is unclear – what about the many other transmissible diseases?

L130 clarify ‘physiological’ fertilization to distinguish ICSI

L145 likely only partially responsible for decreased motility – other factors like decreased mitochondrial function likely also play important roles as you describe in the subsequent paragraph

L171 in what way are the outcomes altered?

L192, L194 references?

L231 sperm are transcriptionally silent, so there is no opportunity to upregulate gene expression

L239 arguably these human studies on sperm methylation are a lot less comprehensive than the animal studies (refs 92-95), as they focus only on subsets of genes

L327, L348 how would the identification of freezability biomarkers be helpful? How would they be practically used in clinics, especially given the ease of other basic semen assessments (particularly motility)

Section 6 – ART outcomes after use of cryopreserved sperm – I think a major point that is missing here is that the effect of cryopreservation is most apparent in low intervention procedures like IUI (e.g. Subak 1992, Am J Obstet Gynec) but less relevant when physiological barriers to sperm transport and fertilization are removed (as in IVF and ICSI)

Reviewer 2 Report

This review is well-planned and conducted. There is an extensive bibliography that denotes great handling of data on the proposed topic. However, there are a few aspects that should be clarified. Sometimes, it is about personal doubts on the subject that should not be corrected in the text.

Comments:

Abstract: If the authors use abbreviations for the first time, they must be explained (ARTs).

Line 78: Authors say: “At present, most ART laboratories use fast thawing at 37°”

Is there a database or reference from which this data has been extracted?

Line 95, 96: The sentence is not understood correctly. Later in the text, the authors say that pathologies such as cancer can make preservation even more difficult. Could you clarify this point or redact again?

Line 108: I have found that there are several unmentioned articles on SARS-Cov2 and cryopreservation of human semen.

 Are they included in the referenced meta-analysis? Include in your review authors with studies with patients outside of Europe. Why not in this case?

Could they not be included because they are not practical in European sperm banks?  Please clarify this or additional references.

Line 159: Authors say:” the initial semen quality, which predicts with high accuracy, sensitivity, and specificity motility recovery after thawing.”   

Could this prediction method be extrapolated to semen from patients without previous pathologies or normozoospermic?

Line 190: Is this a recommendation from the authors or from the WHO guide or another document on cryopreservation?

Line 203: I do not have access to the article with reference 75. I am not clear about the cause of the infertility of these 40 patients. I believe that it is different for the final conclusion  if this infertility is caused by the sperm itself than if it is due to other causes (erection problems, vasectomy...) Could you clarify this point for me?

Line 206: Maybe change the Word “Thorny” to “problematic”

Line 258: by, Rarani and colleagues:  the first comma is not correct.

Line 261: About reference 108. Author says that this effect occurs “immediately after cryopreservation and from 90 days of cryostorage onward” so you can include this conclusion from the author to clarify how the storage time affects chromatin compaction.

Line 350: In my opinion, if the indicated conclusion was reached in reference 125, this paragraph should be considered in section 5 of your review: Possible strategies to prevent the damage and new approaches to sperm cryopreservation.

Line 414: “The Authors”: authors without capitalization

Line 432: “Concerning DNA damage although some studies promote vitrification [159] other authors did not observe differences in post-thaw DNA damage between the two methods”. the commas should be reviewed to better understand the phrase.

Line 458: Considering lyophilization spermatozoa do not preserve viability or motility. If the authors later reference that “as only a single spermatozoon is needed for ICSI of each oocyte, cryopreservation of any live spermatozoon is worthwhile”... Does it not come into contradiction? Please clarify this point or consider redrafting.
